# Development of High-Throughput Multiplex Serology to Detect Serum Antibodies against *Coxiella burnetii*

**DOI:** 10.3390/microorganisms9112373

**Published:** 2021-11-17

**Authors:** Rima Jeske, Larissa Dangel, Leander Sauerbrey, Dimitrios Frangoulidis, Lauren R. Teras, Silke F. Fischer, Tim Waterboer

**Affiliations:** 1Division of Infections and Cancer Epidemiology, German Cancer Research Center (DKFZ), 69120 Heidelberg, Germany; Sauerbrey@stud.uni-heidelberg.de (L.S.); t.waterboer@dkfz.de (T.W.); 2Faculty of Biosciences, Heidelberg University, 69120 Heidelberg, Germany; 3German National Consiliary Laboratory of *Coxiella burnetii*, 70191 Stuttgart, Germany; Larissa.Dangel@rps.bwl.de (L.D.); Silke.Fischer@rps.bwl.de (S.F.F.); 4State Health Office Baden-Württemberg, 70565 Stuttgart, Germany; 5Bundeswehr Institute of Microbiology, 80937 Munich, Germany; dimitriosfrangoulidis@bundeswehr.org; 6Bundeswehr Medical Service Headquarters VI-2, Medical Intelligence & Information (MI2), 80637 Munich, Germany; 7Department of Population Science, American Cancer Society, Atlanta, GA 30303-1002, USA; lauren.teras@cancer.org

**Keywords:** *Coxiella burnetii*, multiplex serology, seroepidemiology, infection marker

## Abstract

The causative agent of Q fever, the bacterium *Coxiella burnetii* (*C. burnetii*), has gained increasing interest due to outbreak events and reports about it being a potential risk factor for the development of lymphomas. In order to conduct large-scale studies for population monitoring and to investigate possible associations more closely, accurate and cost-effective high-throughput assays are highly desired. To address this need, nine *C. burnetii* proteins were expressed as recombinant antigens for multiplex serology. This technique enables the quantitative high-throughput detection of antibodies to multiple antigens simultaneously in a single reaction. Based on a reference group of 76 seropositive and 91 seronegative sera, three antigens were able to detect *C. burnetii* infections. Com1, GroEL, and DnaK achieved specificities of 93%, 69%, and 77% and sensitivities of 64%, 72%, and 47%, respectively. Double positivity to Com1 and GroEL led to a combined specificity of 90% and a sensitivity of 71%. In a subgroup of seropositives with an increased risk for chronic Q fever, the double positivity to these markers reached a specificity of 90% and a sensitivity of 86%. Multiplex serology enables the detection of antibodies against *C. burnetii* and appears well-suited to investigate associations between *C. burnetii* infections and the clinical manifestations in large-scale studies.

## 1. Introduction

The intracellular bacterium *Coxiella burnetii* (*C. burnetii*) is the causative agent of the zoonotic disease Q fever, which mainly affects domestic ruminants. In sheep, goats, and cattle, the infection can cause reproductive failures or late-term abortions. The pathogen can be transmitted by infected ticks or through contact with manure or the birth products of infected animals. Humans in close proximity can also acquire a *C. burnetii* infection, usually by the inhalation of contaminated dust particles [1,2]. Hence, people with occupation-based exposures to infected animals, e.g., veterinarians or farmers, are at risk of developing Q fever. Seroprevalences among these high-risk populations range from 3% to 84%, while seroprevalences among general populations range from 1.3% to 13.6% [2,3,4]. In hyperendemic regions, seroprevalence rates of up to 48.6% were reported [5].

Recently, Q fever has been observed with increasing attention due to several major outbreaks. The largest occurred in the Netherlands between 2007 and 2009 and affected over 4000 people [6,7,8,9]. Reports about a potentially causative role in the development of lymphomas have increased the significance of *C. burnetii* as a public health issue even further [10,11,12,13]. There is particular interest in large-scale studies to examine the role of *C. burnetii*. For this purpose, accurate and cost-effective high-throughput methods are desirable [10,14].

Approximately 50% of the individuals who acquire *C. burnetii* infections develop ‘acute Q fever’ which is accompanied by unspecific, flu-like symptoms and hence often remains misdiagnosed or undiagnosed [1,15,16]. Others do not experience any symptoms at all, which contributes to the underestimation of the actual *C. burnetii* prevalence [17]. In 1–5% of symptomatic and asymptomatic cases, the infection proceeds to a persistent state with severe clinical manifestations, referred to as ‘chronic Q fever’ [18]. Confusingly, this term has been used synonymously with ‘Q fever endocarditis’, which is by far the most frequent clinical manifestation of persistent *C. burnetii* infections. Recent studies have also found *C. burnetii* to play a role in chronic infections of the intima of large arteries [18,19]. Further clinical manifestations include arthritis, osteomyelitis, and hepatitis [1,20,21].

The gold standard for diagnosing Q fever, acute or chronic, is an indirect immunofluorescence assay (IFA). An IFA is a serological detection method based on fixated whole cell *C. burnetii* cultures, either obtained from the spleen of infected mice (Phase I), or after several passages in eggs or cell cultures, causing changes in surface lipopolysaccharides (Phase II).

While acute infections are associated with high anti-Phase II IgM and IgG antibody titers, high levels of anti-Phase I IgG antibodies accompany persistent infections [22]. Anti-Phase I IgG endpoint titers of ≥1:1024 are of special interest in the diagnosis of chronic Q fever. Patients meeting this criterion can be characterized as having ‘possible chronic Q fever’ as proposed by the Dutch consensus guideline on chronic Q fever diagnostics [18,21]. Although there are controversies about the exact cutoff and additional diagnostic criteria, the IFA remains fundamentally important for the diagnosis of Q fever [21,23].

The generation of IFAs, however, requires biosafety level 3 conditions, making it laborious and cost-intensive [22]. Furthermore, the assay conduct is non-automated and, therefore, not suitable for large-scale studies [14]. Hence, multiple approaches to substitute IFAs have been conducted. Antigen ELISAs omit the need to culture *C. burnetii* by using recombinant proteins instead of whole cell lysates and allow the simultaneous analysis of many serum samples.

Here, we present the integration of *C. burnetii* antigens into our multiplex serology platform [24]. This bead-based technique follows the principle of an antigen ELISA, while further enabling the simultaneous measurement of antibodies against multiple antigens in a single reaction. Up to 2000 samples per day can be processed cost effectively, as it has been demonstrated in multiple studies [25,26,27,28]. By including *C. burnetii* antigens, we enable powerful large-scale studies which will help to better understand the role of *C. burnetii* in potential clinical manifestations, e.g., by investigating respective case-control studies. Here, we expressed nine different *C. burnetii* proteins, previously described as immunogenic, for multiplex serology, and tested their ability to discriminate between the sera of *C. burnetii* infected and uninfected individuals.

## 2. Materials and Methods

### 2.1. Reference Sera

Human sera, previously tested for *C. burnetii* infections, were obtained from the German National Consiliary Laboratory of *Coxiella burnetii* in Stuttgart, Germany. The reference serostatus was determined by a semi-quantitative Q fever immunofluorescence assay (IFA), IgG (Focus Diagnostics, Cypress, CA, USA). For each serum, the endpoint titers of *C. burnetii* Phase I and Phase II were measured.

Patients without any detectable antibodies against *C. burnetii* in Phase I or Phase II were considered seronegative. Seropositive patients exhibited endpoint titers against *C. burnetii* Phase I ranging from 1:32 to 1:8192, with a median of 1:512. Endpoints titers against Phase II ranged from 1:64 to 1:65,536, with a median of 1:4096. In total, the reference panel was comprised of 76 sera with positive and 91 sera with negative reference statuses. Of the 76 seropositive references, 28 had a Phase I titer of ≥1:1024.

Since no further information about the sera was available, the seroresponses to five control antigens from four ubiquitous human pathogens were determined (as described below) to compare the two reference groups. These control antigens were the envelope glycoprotein G (gG) from the herpes simplex virus type 1, the DNA polymerase processivity factor BMRF1 (EA-D), the trans-activator protein BZLF1 (ZEBRA) from the Epstein-Barr virus, and the major capsid protein VP1 from the human polyomavirus JC and human polyomavirus 6 [29,30]. Both qualitative (i.e., seroprevalence) and quantitative responses (i.e., antibody titers) for these control antigens were as expected, and no significant differences between the two reference groups were detected in serum dilutions of 1:100 or 1:1000.

### 2.2. The Generation of Recombinant C. burnetii Antigens

Nine *C. burnetii* proteins were selected as antigens for multiplex serology, based on previously reported seroreactivity or expert opinion (Table 1). The respective protein sequences were obtained from the NCBI and were codon-optimized for expression in *Escherichia coli*. Genes were synthesized by Eurofins genomics (Ebersberg, Germany) and cloned into a modified pGEX4T3 vector. Hence, each antigen sequence was expressed as a double fusion protein with an N-terminal GST-tag and a C-terminal peptide tag derived from the SV40 large T-antigen [31]. The successful expression of the full-length recombinant proteins was verified by an anti-tag ELISA, as well as anti-GST and anti-tag Western blot.

### 2.3. Multiplex Serology

Antibody responses to the *C. burnetii* antigens, as well as the control antigens, were determined for each reference serum using multiplex serology, as previously described [20]. Briefly, each GST-fusion protein was coupled to a set of fluorescence-labelled glutathione casein-coated polystyrene beads (SeroMap, Luminex Corp., Austin, TX, USA). The different bead sets were spectrally distinguishable and thereby enabled the simultaneous testing of serum antibodies to different antigens within a single reaction. Serum antibodies, which were bound to the respective antigens, were subsequently quantified using a biotinylated anti-human IgG secondary antibody (1:1000, #109-065-064, Jackson Immunoresearch, West Grove, PA, USA) and streptavidin-R-phycoerythrin (MossBio, Pasadena, MD, USA). The signal of the fluorescent reporter conjugate, as well as the signal assigning the respective antigens, were measured on a Luminex 200 instrument (Luminex Corp., Austin, TX, USA) and were expressed as median fluorescence intensity (MFI).

The reactivity to each antigen was determined in serum dilutions of 1:100 (primary analysis) and 1:1000 (shown in Appendix A and Appendix A). As expected, MFI values were lower in the 1:1000 serum dilution, with many measurements falling below the technical minimum cutoff of 30 MFI (lower limit of quantitation). Nevertheless, the results generated for the 1:100 serum dilution were generally reproduced in the 1:1000 dilution.

### 2.4. Data Processing and Statistical Analysis

Raw MFI values were corrected by subtracting the unspecific background determined in empty controls, as well as the serum-specific anti-GST background. Negative net values were set to 1 MFI. Continuous values were characterized by means and standard deviations. Reference samples with a positive reference status were compared to those with a negative reference status using a nonparametric Mann–Whitney U test. *p*-values below 0.05 were considered statistically significant.

Continuous multiplex serology results were dichotomized using antigen-specific cutoffs obtained by the receiver operating characteristic (ROC) analysis, either by maximizing Youden’s index or by setting a minimum value for the specificity measure. Test agreement was determined by dividing concordant outcomes by the total number of tests.

Cohen’s kappa coefficient was calculated to assess assay agreement beyond chance. Kappa results were interpreted as follows: values <0: less-than-chance agreement; 0–0.2: slight agreement; 0.2–0.4: fair agreement; 0.4–0.6: moderate agreement; 0.6–0.8: substantial agreement; 0.8–1.0: almost perfect agreement.

Analyses were conducted with R 3.6.0 using the package pROC (version 1.16.2) [39].

## 3. Results

### 3.1. Seroresponses to C. burnetii Antigens and Concordance with the Reference Assay

To validate the *C. burnetii* multiplex serology assay, antibody responses to nine selected *C. burnetii* proteins (Table 2) were measured in 167 reference sera, 76 of which were seropositive. The quantitative measurements were visualized according to the predefined serostatus (Figure 1).

MFI values were compared between the two groups (Table 2). Antibody responses against the proteins Com1, CBU_0370, CBU_0937, and GroEL showed a significant difference between the two reference groups. However, only Com1 and GroEL were positively correlated with a positive reference assay serostatus and were, therefore, suitable to detect *C. burnetii* infections.

Sensitivity, specificity, and agreement beyond chance (Cohen’s kappa) were determined and are summarized in Table 3. Moderate agreement with the reference assay was observed for Com1 and GroEL, with kappa values of 0.59 and 0.41, respectively. The achieved specificities and sensitivities were 93% and 64% for Com1, and 69% and 72% for GroEL, respectively. A fair agreement was found for DnaK with a kappa value of 0.25, a specificity of 77%, and a sensitivity of 47%. 

### 3.2. Assay Performance in Patients with High C. burnetii Phase I Endpoint Titers

Considering that especially persistent infections play a role in clinical manifestations, the assay performance is particularly important in samples displaying high levels of anti-Phase I IgG antibodies.

For this purpose, we performed a subgroup analysis of the samples with a positive reference status for *C. burnetii*. Out of the 76 seropositive samples, 28 had an endpoint titer to *C. burnetii* Phase I IgG above the IFA reference assay cutoff of 1:1024. This subgroup was compared to the reference group of 91 seronegative sera without any detectable *C. burnetii* Phase I titers.

In seropositives with high Phase I titers, the performances of Com1, GroEL, and DnaK substantially improved with kappa values of 0.78, 0.56, and 0.40, respectively. The sensitivities were 93%, 84%, and 79% while the specificities reached 86%, 79%, and 68%, respectively (Table 4).

### 3.3. Optimizing Assay Parameters for Seroepidemiological Case-Control Studies

Sensitivity and specificity are statistical measures of test performance which are inversely correlated. Modifying cutoffs increases one of these measures while decreasing the other. Hence, assay cutoffs need to be determined in accordance with the study goals.

Commonly, case-control setups are used to investigate risk factors for clinical outcomes. In these studies, type II errors (false negatives) are preferred over type I errors (false positives) to minimize the risk of chance findings.

Therefore, the cutoffs of the discriminative antigens were pre-specified to yield a minimum specificity of 90% (Table 5). This resulted in a sensitivity of 66% for Com1 and 34% for GroEL. Due to the required minimal cutoff of 30 MFI, a specificity of 90% was not achievable for the antigen DnaK.

A major advantage of multiplexed assays is the possibility to combine antigens to boost assay performance (Figure 2). Considering all the seropositive samples, double positivity to Com1 and GroEL yielded a sensitivity of 71%, which outperformed Com1 as a single marker (Table 5). Combinations that included DnaK were not able to increase assay performance.

In the subgroup of patients with a Phase I endpoint titer of ≥1:1024, Com1, GroEL, and the combination of both achieved sensitivities of 86%, 50%, and 86%, respectively.

By increasing the cutoffs of Com1 and GroEL from 100 and 110 MFI, respectively, to 180 and 400 MFI, respectively, a specificity of 100% was achieved at a marginal loss in sensitivity (from 86% to 79%).

## 4. Discussion

We demonstrated that multiplex serology is a valuable tool for large-scale studies addressing *C. burnetii* infections. Analyzing up to 2000 serum samples per day, this technique has been used in diverse seroepidemiological studies to detect antibodies against a variety of different pathogens, e.g., *Helicobacter pylori*, human papillomaviruses, and herpes viruses [25,26,27,28]. The inclusion of *C. burnetii* broadens the spectrum of measurable pathogens and enables powerful studies to investigate the association between *C. burnetii* infections and potential clinical outcomes.

The best discrimination between seropositive and seronegative *C. burnetii* reference samples was achieved by the outer membrane-associated protein Com1 [40]. Its immunogenicity was initially described in 1990 and has been reproduced in several studies. Sensitivities range from 47% to 94% and specificities range from 68% to 90%, varying by the immunoassay method, host species, and the investigated population [36,38,41,42]. To the best of our knowledge, the sensitivity and specificity of our combined antigen assay is the highest reported in humans so far.

Especially in chronic Q fever patients, Com1 achieved a good performance [33,43,44]. Vranakis et al. reported a specificity and sensitivity above 90%. Due to a lack of clinical or follow-up data from our reference groups, we were not able to diagnose and analyze chronic Q fever cases separately. Instead, we performed a subgroup analysis based on the endpoint titer against *C. burnetii* Phase I, which is strongly associated with chronic Q fever [45].

As proposed by the Dutch Q fever consensus group, patients with an IFA Phase I endpoint titer of ≥1:1024 were considered ‘possible chronic Q fever’ cases [18,21]. This definition has been controversially discussed in a diagnostic context [23]. Epidemiologically, it is, however, suited to subdividing patients by their risk of developing chronic Q fever. In this subgroup, Com1 achieved a sensitivity of 86% and specificity of 93%. We thereby demonstrate that multiplex serology performs particularly well in patients with a higher risk of chronic Q fever and the respective clinical manifestations.

Com1 was described as a good serological marker by Xiong et al. [32]. Their study showed that among seven potential molecular markers, Com1 antibodies were particularly persistent, and were detected in 52% of acute late Q fever patients and 50% of convalescent Q fever patients. This finding strongly supports the role of Com1 as a serological biomarker, as clinical manifestations, e.g., endocarditis or hepatitis, might occur years after the primary infection. Especially in large-scale settings, biomarkers which are detectable within a large time frame are of special advantage, as large heterogeneity between patient samples might exist, e.g., due to different time spans between the blood draw and the primary infection.

GroEL and DnaK were also persistent and accurate molecular markers described by Xiong et al. [32,37,44,46,47,48]. Using multiplex serology, they achieved specificities of 84% and 79%, respectively, and sensitivities of 79% and 68%, respectively.

GroEL is a heat shock protein utilized as serological marker in a broad range of different organisms, e.g., *Burkholderia pseudomallei*, and *Helicobacter pylori* [49,50]. It does not come as a surprise that cross-reactivities with *C. burnetii* GroEL have been reported, especially in cases of rickettsial spotted fever and Legionella pneumonia infections [32]. This might explain the relatively high proportion of false positives when GroEL was used as a sole marker. The same principle applies to DnaK, which is also a chaperone used as serological marker for multiple organisms [51,52]. The reference sera tested here derived from individuals exhibiting unspecific flu-like symptoms, which were forwarded to the reference laboratory to be tested for *C. burnetii* infections. In the individuals who already showed signs of infection, the false positive rates might, therefore, be elevated compared to an asymptomatic population.

A main advantage of multiplex serology is, however, that unspecific reactivity can be addressed by considering multiple antigens. Here, GroEL can be combined with Com1 measurements to increase the specificity. When optimizing assay parameters for case-control studies, we pre-specified specificity at 90%. Among all 76 seropositive samples, sensitivity achieved 71%, which outperformed Com1 as a sole marker. Considering the patients with the high Phase I titer only, the sensitivity reached 86%. Alternatively, the specificity can be set to 100% by increasing the cutoffs accordingly, which can be of special interest to investigate risk factors to rare outcomes, e.g., lymphomas. The assay performance in these kind of studies, case-control or population-based, needs to be validated further in the future.

Overall, the integration of *C. burnetii* into the multiplex serology platform enables large-scale seroepidemiological studies to investigate possible associations with this infection. Multiplexing *C. burnetii* antigens with antigens derived from other pathogens will enable the assessment of the seroreactivity to multiple infectious agents simultaneously, providing a broader view on different infections and co-infections.

## 5. Summary

This study presents the development of a multiplex serology assay for the high-throughput detection of *C. burnetii* antibodies in human sera. Nine selected antigens (YbgF, CBU_937, CBU_370, AdaA, Com1, GroEL, DnaK, Mip, and CBU_1425) were expressed as recombinant GST-fusion proteins and were used as targets in this bead-based suspension array. Validation was performed in a panel of 167 reference sera from whom 91 were pre-classified as seronegative and 76 as seropositive. Com1, GroEL, and DnaK were able to discriminate between these two groups with specificities of 93%, 69%, and 77%, respectively, and sensitivities of 64%, 72%, and 47%, respectively. A combination of the markers Com1 and GroEL outperformed single antigens with a combined specificity and sensitivity of 90% and 71%, respectively. By restricting the analysis to a subgroup of seropositives exhibiting an anti-Phase I endpoint titer of ≥ 1:1024, the specificities of Com1 and GroEL increased to 93% and 84%, respectively, and the sensitivities to 86% and 79%, respectively. Combining both markers resulted in a specificity of 90% and sensitivity of 86%.

The multiplex serology assay presented here enables large-scale seroepidemiological studies to investigate the associations between *C. burnetii* infections and the clinical manifestations.

## Figures and Tables

**Figure 1 microorganisms-09-02373-f001:**
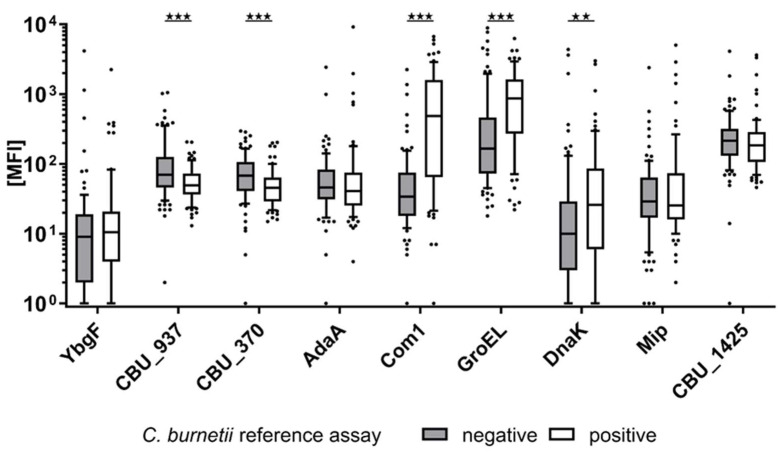
Antibody levels in the reference panel of 76 *C. burnetii* seropositive and 91 seronegative reference samples. CBU_937, CBU_370, Com1, and GroEL show a significantly different distribution (*p* < 0.05) between the two reference groups. Stars above boxplots indicate the level of significance, *p* < 0.01 (**); *p* < 0.01 (***).

**Figure 2 microorganisms-09-02373-f002:**
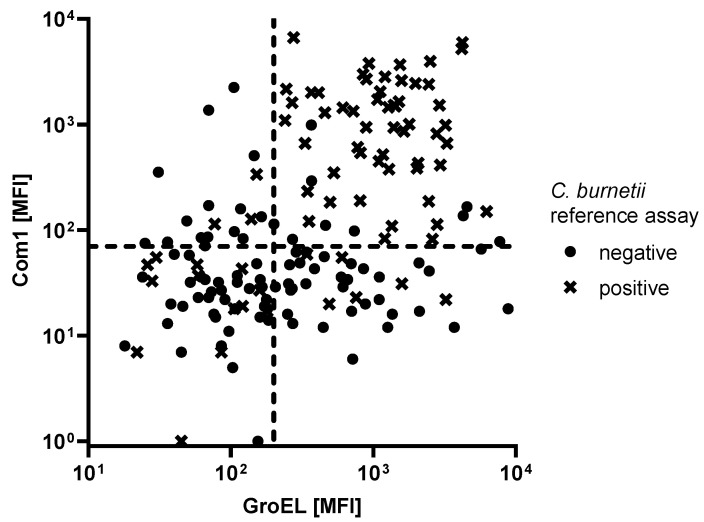
Antibody responses against the *C. burnetii* antigens Com1 and GroEL measured in 76 *C. burnetii* seropositive and 91 seronegative patients. Dashed lines (x = 200 MFI; y = 70 MFI) indicate the respective cutoffs to obtain a specificity of 90% and a combined sensitivity of 71%.

**Table 1 microorganisms-09-02373-t001:** *C. burnetii* proteins selected as target antigens for the multiplex serology assay.

Antigen Locus Tag	Antigen Symbol	RefSeq Accession	Protein Description	Subcellular Location	Reference
CBU_0092	YbgF	NP_819144.2	Cell division coordinator CpoB	Periplasm	Xiong et al. [32]
CBU_0937		NP_819950.2	LbtU family protein (siderophore porin)	Secreted	Sekeyova et al. [33]
CBU_0370		NP_819410.1	Membrane-associated protein	Membrane	
CBU_0952	AdaA	NP_819961.1	Acute disease antigen A	Secreted	Coleman et al. [34] Chen et al. [35]
CBU_1910	Com1	NP_820887.1	Outer membrane protein	Membrane	Stellfeld et al. [36] Jiao et al. [37] Xiong et al. [32] Chen et al. [35] Beare et al. [38]
CBU_1718	GroEL	NP_820699.1	60 kDa chaperonin	Cytoplasm	Xiong et al. [32] Coleman et al. [34]
CBU_1290	DnaK	NP_820282.1	Chaperone	Cytoplasm, Endoplasm, Membrane	Xiong et al. [32]
CBU_0630	Mip	NP_819660.1	Peptidyl-prolyl cis-trans isomerase	Secreted	Xiong et al. [32]
CBU_1425		NP_820409.1	17 kDa common antigen	Membrane	

**Table 2 microorganisms-09-02373-t002:** Quantitative seroresponses to nine selected *C. burnetii* antigens measured with multiplex serology in a serum dilution of 1:100. Reference samples were grouped according to a positive (pos.) or negative (neg.) *C. burnetii* reference assay test result. Groups were compared using Mann–Whitney U test. A *p*-value below 0.05 was considered statistically significant.

Antigen Locus Tag	Antigen Symbol	*C. burnetii* Pos. (MFI) (SD)	*C. burnetii* Neg. (MFI) (SD)	*p*-Value
CBU_0092	YbgF	65 (266)	77 (451)	0.58
CBU_0937		60 (38)	128 (173)	<0.001
CBU_0370		55 (40)	84 (60)	<0.001
CBU_0952	AdaA	228 (1073)	100 (269)	0.26
CBU_1910	Com1	1094 (1423)	105 (291)	<0.001
CBU_1718	GroEL	1209 (1205)	697 (1518)	<0.001
CBU_1290	DnaK	151 (470)	138 (621)	<0.01
CBU_0630	Mip	206 (703)	81 (259)	0.81
CBU_1425		324 (584)	313 (469)	0.13

**Table 3 microorganisms-09-02373-t003:** Performance of nine *C. burnetii* antigens measured in 76 reference samples with positive, and 91 with negative, *C. burnetii* serostatuses. Quantitative measurement values were dichotomized using the given cutoff (maximum value of the Youden’s index) to calculate assay specificity, sensitivity, and agreement (kappa).

Antigen Locus Tag	Antigen Symbol	Cutoff (MFI)	Specificity (%)	Sensitivity (%)	Cohen’s Kappa κ (95% CI)
CBU_0092	YbgF	32	89	21	0.11 (−0.05 to 0.27)
CBU_0937		67	54	72	−0.26 (−0.41 to −0.11)
CBU_0370		67	53	80	−0.32 (−0.47 to −0.18)
CBU_0952	AdaA	33	74	42	−0.15 (−0.29 to −0.01)
CBU_1910	Com1	178	93	64	0.59 (0.47 to 0.72)
CBU_1718	GroEL	340	69	72	0.41 (0.27 to 0.55)
CBU_1290	DnaK	30 ^1^	77	47	0.25 (0.10 to 0.40)
CBU_0630	Mip	30 ^1^	51	47	−0.02 (−0.17 to 0.13)
CBU_1425		124	79	38	−0.15 (−0.30 to −0.01)

^1^ A technical minimum cutoff of 30 MFI was applied.

**Table 4 microorganisms-09-02373-t004:** Quantitative seroresponses to nine selected *C. burnetii* antigens measured with multiplex serology in a serum dilution of 1:100. Reference samples were grouped according to a positive (pos.) or negative (neg.) *C. burnetii* reference assay test result. Groups were compared using Mann–Whitney U test. A *p*-value below 0.05 was considered statistically significant.

Antigen Locus Tag	Antigen Symbol	Cutoff (MFI)	Specificity (%)	Sensitivity (%)	Cohen’s Kappa κ (95% CI)
CBU_0092	YbgF	30 ^1^	86	25	0.12 (−0.13 to 0.37)
CBU_0937		82	42	93	−0.30 (−0.52 to −0.08)
CBU_0370		58	60	82	−0.31 (−0.48 to −0.14)
CBU_0952	AdaA	43	63	50	−0.08 (−0.24 to 0.08)
CBU_1910	Com1	179	93	86	0.78 (0.64 to 0.91)
CBU_1718	GroEL	747	84	79	0.56 (0.39 to 0.73)
CBU_1290	DnaK	31	79	68	0.40 (0.21 to 0.59)
CBU_0630	Mip	30 ^1^	51	43	−0.05 (−0.23 to 0.14)
CBU_1425		124	79	43	−0.10 (−0.24 to 0.03)

^1^ A technical minimum cutoff of 30 MFI was applied.

**Table 5 microorganisms-09-02373-t005:** Final multiplex serology assay characteristics and performance of antigens Com1 and GroEL, as well as a combination of both antigens. Assay performance was determined in an overall *C. burnetii* positive population (n = 76) and a subgroup with high Phase I endpoint titers of ≥1:1024, as an approximation for chronic Q fever (n = 28).

Seropositive Reference Population	Antigen	Cutoff (MFI)	Specificity (%)	Sensitivity (%)
76 *C. burnetii* seropositive patients	Com1 (CBU_1910)	140	90	66
	GroEL (CBU_1718)	1360	90	34
	Com 1 + GroEL (double positives)	70/200	90	71
28 *C. burnetii* seropositive patients with a Phase I titer of ≥1:1024	Com1 (CBU_1910)	140	90	86
	GroEL (CBU_1718)	1360	90	50
	Com 1 + GroEL (double positives)	100/110	90	86
		180/400	100	79

## Data Availability

Data is available upon request.

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
