# Peer review of "Development of High-Throughput Multiplex Serology to Detect Serum Antibodies against Coxiella burnetii"

_microorganisms, 2021, doi:10.3390/microorganisms9112373_

Round 1

Reviewer 1 Report

The manuscript “High-throughput Multiplex Serology to Detect Serum Antibodies against Coxiella burnetii” describes the development of a multiplex ELISA test for the diagnosis of Q fever in humans.

The topic is worthy of investigation due to public health implications.

The study is well-conducted and scientifically sound.  

Therefore, I do recommend acceptance for publication in “Microorganisms” following some minor revisions.

Please, pay attention to form and style, i.e. the use of italics for scientific names.

Line 33: Correct “agents” into the singular form “agent”.

In the Introduction, you should give some brief information on the transmission modes of Coxiella burnetii in humans. Besides, you should state that this is a zoonosis, and animals get infected by ticks, since these arthropods represent a reservoir of the pathogen. A mention of the reproductive problems in ruminants should be included. Appropriate references should be added in this regard. 

Moreover, you should add some seroprevalence data at the global level and in Europe. Appropriate references should be added in this regard. 

A conclusion paragraph summarizing the main findings of the study should be provided.

Author Response

We would like to thank the reviewer for taking the time to review the manuscript and provide us with useful comments and thoughts. 

- Line 33: Correct “agents” into the singular form “agent”.

'Agents' was corrected to 'agent'.

- In the Introduction, you should give some brief information on the transmission modes of Coxiella burnetii in humans. Besides, you should state that this is a zoonosis, and animals get infected by ticks, since these arthropods represent a reservoir of the pathogen. A mention of the reproductive problems in ruminants should be included. Appropriate references should be added in this regard. 

Thank you for pointing this out. We revised the introduction and added the missing information.

- Moreover, you should add some seroprevalence data at the global level and in Europe. Appropriate references should be added in this regard. 

Appropriate information on the seroprevelance of C. burnetii was added to the introduction.

-  A conclusion paragraph summarizing the main findings of the study should be provided.

We added a paragraph '6. Summary' and summed up the main assay parameters.

Reviewer 2 Report

The manuscript by Jeske R. et al. entitled “High-throughput Multiplex Serology to Detect Serum Antibodies against Coxiella burnetii” describes the results of a multiplex serological study using IFA enabling the simultaneous measurement of nine antibodies against multiple antigens in a single reaction. However, the total number of analyzed sera (76 seropositive + 91 seronegative) are few, this manuscript present important results. The concept is excellent, the manuscript is well written. The MS needs minor revision.

Few comments:

L46: Revise the sentence, ‘1-5% of symptomatic and asymptomatic cases…’

L233-239: revise the paragraph. This paragraph rather presents the results of the study,  and is more a part of the results section. Some information is repeated, e.g.  ‘provided by the German National Consiliary Laboratory’.

L241: The sentence is confusing, ‘the sensitivities of Com1, GroEL and the combination of both were increased to 86%, 79% and 86%’. Increased from what %?

Author Response

We would like to thank the reviewer for taking the time to review the manuscript and provide us with useful comments and thoughts. 

- Point 1 L46: : Revise the sentence, ‘1-5% of symptomatic and asymptomatic cases…

We revised the respective sentence in L46 as proposed.

- L233-239: revise the paragraph. This paragraph rather presents the results of the study,  and is more a part of the results section. Some information is repeated, e.g.  ‘provided by the German National Consiliary Laboratory’.

This paragraph was shortened and moved to '6. Summary'

- L241: The sentence is confusing, ‘the sensitivities of Com1, GroEL and the combination of both were increased to 86%, 79% and 86%’. Increased from what %?

This section was rephrased. It is now more clear, that these measures refer to a subgroup among the seropositives (high Phase I), as opposed to all seropositives. As stated above, this sentence is now part of the summary.